# Enamel Demineralization Resistance and Remineralization by Various Fluoride-Releasing Dental Restorative Materials

**DOI:** 10.3390/ma14164554

**Published:** 2021-08-13

**Authors:** Min-Ji Kim, Myung-Jin Lee, Kwang-Mahn Kim, Song-Yi Yang, Ji-Young Seo, Sung-Hwan Choi, Jae-Sung Kwon

**Affiliations:** 1Department and Research Institute of Dental Biomaterials and Bioengineering, Yonsei University College of Dentistry, Seoul 03722, Korea; lovely5171@hanmail.net (M.-J.K.); kmkim@yuhs.ac (K.-M.K.); syyang88@yuhs.ac (S.-Y.Y.); 2BK21 PLUS Project, Yonsei University College of Dentistry, Seoul 03772, Korea; 3Division of Health Science, Department of Dental Hygiene, Baekseok University, Cheonan 31065, Korea; dh.mjlee@bu.ac.kr; 4Department of Orthodontics, Institute of Craniofacial Deformity, Yonsei University College of Dentistry, Seoul 03772, Korea; jyseo13@yuhs.ac

**Keywords:** fluoride-releasing restorative material, acid neutralizing property, fluoride release, calcium concentration, demineralization resistance, remineralization

## Abstract

The aim of this study is to investigate the resistance of various fluoride-releasing restorative materials against the demineralization and remineralization of enamel surfaces, including those that have been recently introduced to the market. Three different fluoride-releasing restorative materials were considered: glass ionomer (FI), resin-modified glass ionomer (RL), and an alkasite restorative material (CN). The acid neutralization ability was investigated using pH measurement, and the concentrations of released fluoride and calcium ions were measured. Finally, the demineralization resistance and remineralization effects of enamel were observed using a microhardness tester and SEM. CN showed an initial substantial increase in pH followed by a steady increase, with values higher than those of the other groups (*p* < 0.05). All three groups released fluoride ions, and the CN group released more calcium ions than the other groups (*p* < 0.05). In the acid resistance test, from the microhardness and SEM images, the CN group showed effective resistance to demineralization. In the remineralization test, the microhardness results showed that the FI and CN groups recovered the microhardness from the values of the demineralized enamel surface (*p* < 0.05). This was confirmed by the SEM images from remineralization tests; the CN group showed a recovered demineralized surface when immersed in artificial saliva for 7 days. In conclusion, alkasite restorative material can be an effective material when used in cariogenic environments.

## 1. Introduction

Chemically, enamel is hydroxyapatite (HA); over 90 wt % of its weight is composed of minerals such as calcium and phosphate, which are critical determinants of the physicochemical properties of enamel [1]. According to Miller’s chemico-parasitic theory, when oral bacteria metabolize sugar from dietary carbohydrates, the drop in pH due to acid production induces demineralization, and a continuous process of demineralization ultimately causes dental caries [2]. Thus, demineralization is an important factor for the advancement in dental caries, but can be prevented by lowering the pH by releasing high levels of ion in an acid environment for the exchange of hydrogen ions, which, in turn, neutralizes the acid [3].

According to a previous study, pH ≥ 6.0 is a safe zone with a low risk of dental caries, pH 5.5–6.0 is the zone that is potentially cariogenic, and pH 4.0–5.5 is the zone with a high risk of dental caries incidence [4]. Therefore, an ideal restorative material with the ability to resist demineralization should rapidly neutralize the pH by ion release, so the surrounding enamel is not vulnerable to acid exposure after dental restoration treatment.

Remineralization is the restoration of the demineralized tooth through mineral re-deposition in enamel deficient in calcium and phosphate [5]. When saliva is supersaturated with minerals such as calcium in the oral environment, remineralization is accelerated, and fluoride promotes enamel remineralization by absorbing the demineralized surface of enamel and the deposits of calcium ions in the tooth [6]. For remineralization against initial dental caries, the original HA structure is restored when pH is neutralized through saliva and the supply of calcium and phosphate [7]. However, in a highly acidic environment, this natural remineralization is insufficient for achieving enamel remineralization. Thus, various restorative materials providing an enamel-remineralization effect have recently been developed, including adhesive-containing nanoparticles of amorphous calcium phosphate (NACP) [8]. Fluoride-releasing restorative materials exhibit resistance to enamel demineralization, providing an enamel demineralization effect when applied to the tooth [9,10].

Glass ionomer consists of acrylic acid alumino-fluoro-silicate glass and others, and its hardening is mediated via the acid–base reaction [11]. This material provides an enamel demineralization resistance effect based on fluoride release [12]. Although it can release and recharge fluoride, the disadvantage is that the material dissolves cations and anions upon contact with moisture prior to completely hardening [13]. As a solution to this problem, resin-modified glass ionomer was developed, which uses the methacrylate polymerization reaction in addition to the acid-base reaction, thereby overcoming the moisture sensitivity problem of glass ionomer [14]. Nevertheless, its mechanical strength is relatively lower than that of composite resin [15].

One of the newly developed restorative materials known as an “alkasite” is a dual-cured, resin-based material containing alkasite glass, which is alkaline glass containing CaF_2_, SiO_2_, CaO, and Na_2_O, forming OH^−^ groups upon reaction with water and releasing Ca^2+^ to result in acid neutralization and remineralization [3]. One of the examples of the alkasite material is Cention N (CN), which is available on the market, with limited information in terms of acid neutralization and remineralization. This material also releases fluoride and the manufacturer asserts that it can be used as an alternative to amalgam [16]. In conclusion, CN has an alkaline glass filler embedded into a resin matrix and releases fluoride; this feature simultaneously provides those of the two previously mentioned materials. However, to date, no study has evaluated the various fluoride-releasing restorative materials by simulating the clinical conditions of dental caries based on an enamel model. Therefore, the aim of this study was to compare the demineralization resistance and remineralization effects of various fluoride-releasing restorative materials.

## 2. Materials and Methods

### 2.1. Materials

Three types of fluoride-releasing restorative materials that are commercially available on the market were considered in this study. The selected materials have different types and compositions, which are listed in Table 1.

### 2.2. Scanning Electron Microscopy (SEM) Observation of Materials’ Surfaces

All the specimens were prepared using a stainless-steel mold with a 10 mm diameter and 2 mm thickness. The specimens were subsequently removed from the mold, and the surface was polished with 800 grit abrasive paper using a water-cooled rotating polishing machine (Ecomet 30, Buehler Ltd., Lake Bluff, IL, USA). The surface was analyzed with one specimen from each group. The specimen surface was sputter-coated with platinum under a vacuum evaporator before analysis, and observed using scanning electron microscopy (SEM; S-300N, Hitachi Ltd., Nagoya, Japan) at 2000× magnification with an accelerating voltage of 15 kV.

### 2.3. Acid Neutralization Property

To fabricate the specimens, a stainless steel mold (25 mm × 2 mm × 2 mm), polyester film, and microscope glass slide (76 mm × 26 mm × 1 mm) were used. All materials were manufactured according to the manufacturers’ instructions. The stainless-steel mold placed on poly-ester film was placed in microscope glass slide. Then, the mold was filled with FI mixture and covered with polyester film and microscope glass, and the mold was pressed between two opposing microscope glass slides. The RL capsules were homogenized using a high-power mixer for 10 s. The stainless-steel mold on polyester film was placed in microscope glass slide. Then, the mold was filled with RL material. Each side of the specimen was photocured using a light curing unit (Elipar^TM^ S10, 3M ESPE Co., Seefeld, Germany) for 20 s. The CN mixture was placed into the mold and covered with a polyester film and microscope glass. Then, the irradiation procedure was repeated using the same method as for RL. The excess materials of all specimens were removed with 400, 800, and 1200 grit abrasive paper. To investigate acid neutralization, three specimens of each fluoride-releasing restorative material were immersed in 2.14 mL of lactic acid solution (pH 4.0), yielding a specimen volume/acid solution ratio of 0.14 cm^3^/1 mL [12], at a temperature of 25 ± 1 °C. A digital pH meter (Orion 4 Star, Thermo Fisher Scientific Inc., Singapore) was used for measurement. The pH was recorded every minute and the total measurement time was 100 min.

### 2.4. Fluoride Ion Release

The specimens were fabricated using a stainless-steel mold with a 10 mm diameter and 1 mm thickness [13]. A total of 6 specimens of each material were produced. The prepared disk specimens were polished with 400, 800, and 1200 grit abrasive paper. Each specimen was immersed in 5 mL distilled water and the solution was not changed during the test period. Seven days after the start of the immersion, 10 vol % of TISAB Ⅲ buffer was added to stabilize the fluoride solution. Fluoride ion release from the specimens was calculated using a pH/ion meter (Orion 4 Star pH/ISE Benchtop, Agilent, Thermo Fisher Scientific Inc., Singapore).

### 2.5. Calcium Concentration

The specimens were fabricated using a stainless-steel mold with a 10 mm diameter and 2 mm thickness. Five disk-shaped specimens were prepared for the study duration period. The specimens were polished with 400, 800, and 1200 grit abrasive paper. After polishing, each specimen was immersed in 7 mL distilled water and stored in a 37 °C incubator. After 7, 15, and 30 days from the start of the immersion, the calcium release from the specimens was calculated using inductively coupled plasma-mass spectrometry (ICP-MS) (Agilent 7900, Agilent, Santa Clara, CA, USA). Three calibration curves were prepared from the standard stock solutions to determine the calcium ion concentration.

### 2.6. Demineralization Resistance Test

Extracted sound bovine teeth were used in this study. Separated crowns were embedded in acrylic resin using a Teflon mold. The embedded bovine teeth were polished with 400, 800, and 1200 grit abrasive paper. After polishing, an artificial cavity was formed in the center of the enamel. Then, the cavity was filled with each restorative material. The demineralization solution consisted of 2.2 mM CaCl_2_, 2.2 mM NH_2_PO_4_, and 50 mM acetic acid. Then, the demineralization solution was adjusted to pH 4.4 using 1 M KOH [14]. The microhardness of the enamel surface was measured in two steps: before demineralization (baseline) and after immersion in demineralization solution. The measurements were recorded using a Vickers hardness tester (Model DMH 2, Matsuzawa Seiki Co., Tokyo, Japan) with a 100 g load for 10 s of dwell time. The enamel was also observed using SEM (JEOL-7800F, JEOL Ltd., Tokyo, Japan) with an accelerating voltage of 10 kV at 10,000× magnification.

### 2.7. Remineralization Test

The specimens were prepared using the same method as described for the demineralization resistance test. After the demineralization procedure, the cavity-formed tooth was filled with restorative materials. Then, the specimens were immersed in artificial saliva (AS), which was fabricated in accordance with ANSI/ADA specification No. 41 (recommended standard practices for biological evaluation of dental materials). The microhardness of the enamel surface was determined as above. The enamel surface was measured in three steps: sound enamel (baseline), after immersion in demineralization solution, and following immersion in artificial saliva. SEM (JEOL-7800F, JEOL Ltd., Tokyo, Japan) images were also obtained with accelerating voltages of 5, 10, and 15 kV at 10,000× magnification.

### 2.8. Statistical Analysis

The statistical analyses were carried out by one-way ANOVA followed by Tukey’s post hoc test. Additional analyses were performed for the results of the microhardness and demineralization resistance test using an independent *t*-test. All statistical analyses were performed using the SPSS 23 software program (IBM Corp, Armonk, NY, USA) and the level of significance was fixed at 0.05.

## 3. Results

### 3.1. Scanning Electron Microscopy (SEM) Images of the Materials’ Surfaces

SEM images of the materials’ surfaces are shown in Figure 1. The surface of the FI group presented many voids and cracks compared to the other groups. The RL surface was rougher than that of CN, containing small pores on the surface. The CN group showed an irregular surface, with the presence of visible particles, which were approximately 20–30 µm in size.

### 3.2. Acid Neutralizing Property

Figure 2 represents the pH change following the immersion of specimens in lactic acid solution (pH 4.0) for 100 min. The curve of both FI and RL show minimal changes in pH with the final pH at 100 min being 4.75 ± 0.92 and 3.94 ± 0.10, respectively. In contrast, the pH of CN increased rapidly during the first 30 min, where pH increased above the critical value of 5.5. The steady increase in pH in the CN group finally reached pH 9.15 ± 0.25 following 100 min of immersion. The final pH values at 100 min showed significant differences among the three groups (*p* < 0.05).

### 3.3. Fluoride Ion Release

Figure 3 depicts the results of fluoride ion release from each experimental material for 7 days. According to the results, the FI and RL groups were not significantly different in the amount of fluoride release (*p* > 0.05). However, a significant difference was observed between these two groups and the CN group (*p* < 0.05).

### 3.4. Calcium Concentration

The results of the calcium ion concentration are shown in Table 2. The CN group released more calcium ions than the other groups in all periods (*p* < 0.05). Furthermore, all groups showed no significant differences in calcium ion release in each period (*p* > 0.05).

### 3.5. Demineralization Resistance Test

The results show that the microhardness of the enamel in the FI and RL groups was significantly lower after compared to before immersion in demineralization solution (*p* < 0.05; Figure 4A). However, we found no significant changes in the hardness values for CN before and after the immersion. Representative SEM images of an enamel surface restored with the experimental restorative materials are shown in Figure 4B. The demineralized enamel surface showed a rough and uneven surface, unlike the sound enamel. Comparing the demineralized enamel surfaces, all groups showed a smoother surface than the base enamel immersed in demineralization solution. However, the surfaces of the CN group showed less demineralization compared to the other groups.

### 3.6. Remineralization Test

The microhardness significantly decreased after demineralization treatment compared to sound enamel in all the groups (*p* < 0.05; Figure 5A). The FI group did not show recovery of hardness compared to the sound enamel before demineralization, but at 30 days, it recovered with a minimal increase in the microhardness value (*p* < 0.05). The RL group did not recover to its previous level after demineralization treatment for all periods. However, after immersion in the artificial saliva, only the CN group showed microhardness values similar to those of sound enamel for all periods (*p* > 0.05). Representative SEM images of all groups are shown in Figure 5B. Compared to the demineralized enamel surface, FI and RL groups showed a smooth enamel surface after 7 days. However, the CN group showed a smoother surface than the other groups after 7 days, whereas at 15 and 30 days, the enamel appeared to be irregular on the surface.

## 4. Discussion

This study was conducted to evaluate the properties of different fluoride-releasing restorative materials, including a recently developed material known as ‘alkasite’ (CN). Additionally, their demineralization resistances and remineralization effects were compared. As far as we are concern, this is the first paper that considered such properties of alkasite material and other fluoride-releasing restorative material, particularly under clinically similar environments.

In the acid neutralization test, the change in pH was monitored after immersing each fluoride-restorative material in lactic acid solution at pH 4.4. The CN group showed a level higher than the critical pH 5.5 within 30 min. According to a previous study, the drop in pH due to acid produced by pathogenic bacteria induces enamel demineralization, and at a pH lower than the critical pH 5.5, enamel minerals dissolution was observed [17]. Based on the Stephan curve, the pH, after rising, decreased to the high-risk cariogenic zone between 5 and 20 min, and the initial pH was generally restored within 30–60 min [18]. It is thus critical that the pH is increased within the shortest time to avoid pH 4.0, where the risk of dental caries is high. As shown in Figure 2, the pH change observed at pH 4.0 was higher than at the critical pH 5.5 for the CN group within 30 min, as the pH in the group increased more rapidly than in other groups within the first 10 min, which is when the pH fell to the lowest level following acid exposure. However, the other two groups showed no significant change in pH. The CN group containing CaO and Na_2_O alkaline glass can create conditions for acid neutralization based on the alkali ions as OH^−^ groups are released in aqueous environments. Na^+^ is one of the main ions enabling the exchange process with protons (H^+^) from the solution; the Na_2_O content determines the alkalizing ability of the filler [19]. By containing an alkaline glass, the CN group has the capacity for rapid acid neutralization, which confirms the effect of demineralization resistance when the restorative material is applied to the enamel. In addition, the fluoride-ion release forms fluoroapatite, which has a more effective demineralization resistance than hydroxyapatite [20]. Consequently, CN was found to have superior demineralization resistance than the other two groups by simultaneously releasing various ions.

The microhardness of the enamel surface can be non-destructively measured, and is related to the mineral content of the enamel, while reflecting the mechanical properties of the tooth structure. The method has also been applied in studies that quantitatively estimated the initial enamel demineralization. For a cariogenic case, it is suitable for examining the demineralization after acid challenge [21,22,23]. As shown in Figure 4A, the FI and RL groups displayed significantly different decreases in microhardness following demineralization when compared to the sound enamel before demineralization. In contrast, the CN group showed no significant difference in the acid-challenged enamel compared to the sound enamel. This finding is likely due to the demineralization resistance capacity of the CN group in an acidic environment, which arises from the acid neutralization capacity based on the release of OH^−^ groups. In a previous study, during the process of demineralization at pH 4.0, restorative materials with acid neutralization capacity simultaneously exhibited a demineralization resistance effect [24]. In addition, CN groups release a higher level of calcium ions at pH 4.0 artificial saliva than at pH 7.3 Tris buffer [3]. A previous study reported that when the pH decreases, enamel demineralization relies on the level of calcium saturation [25]. Consequently, the CN group was predicted to be a restorative material with a capacity for initial acid neutralization, as well as providing an demineralization resistance effect through calcium ion release.

SEM observation is a method widely used for the qualitative estimation of demineralization. SEM images also provide information about surface properties [26,27]. In a previous study, microhardness determination and SEM image analysis were carried out to evaluate the demineralization resistance of fluoride-releasing materials with demineralized enamel [28]. In this study, the SEM images were examined in addition to the enamel surface microhardness, and the result showed that in terms of demineralized enamel surface, the CN group had the smoothest surface due to the reduced influence of the demineralization process compared to the other groups.

During the remineralization process, calcium and phosphate ions penetrate the demineralized pores, thereby restoring the microhardness of enamel, which is referred to as a re-hardening process. This process is also the result of mineral precipitation. According to previous studies, composite resin containing amorphous calcium phosphate (ACP) is known to exhibit a remineralization effect as it releases high levels of calcium and phosphate ions [29]. CN releases various ions such as Ca^2+^, Na^+^, and F^−^, which increase the pH and form apatite when immersed in artificial saliva [30]. As shown in Table 2, the CN group released calcium ions; when the hardness of demineralized enamel was compared, significant differences were observed at days 7, 15, and 30, presumably due to the remineralization effect based on calcium release. The FI group showed a significantly increased hardness of the enamel surface on day 30. The explanation for this can be found in a previous study, where microhardness was reported to be restored upon immersing demineralized enamel in artificial saliva, which was ultimately due to the influence of the artificial saliva [31]. These results are supposed to be affected by the formation of apatite through ion release.

When observing the remineralized enamel surface using SEM, the CN group showed a smoother enamel surface than the other groups, which was expected because it is a material that simultaneously releases fluoride and calcium ions. In a previous study, effective remineralization of subsurface enamel was found when materials containing calcium as well as fluoride ions were applied to human enamel [32]. No significant changes in the FI and RL groups with each a period were observed. In contrast, the CN group showed irregular and thin pieces of enamel surface on days 15 and 30, which was expected to be influenced by the material when observing the material surface. Consequently, the CN group is expected to have a remineralization effect for 7 days. To exactly verify the caries progression, SEM images and microhardness of enamel surface have limitations in evaluating enamel subsurface lesions. However, this measurement is useful for evaluating initial structural changes according to acid exposure and applied materials.

This study confirmed for the first time that the tested alkasite restorative material (CN) showed superior fluoride releasing ability than other fluoride-releasing restorative materials, resulting in quicker acid neutralization and remineralization. Still, the study was limited to the test conditions considered in this study as well as one type of commercially available material, where further study, including animal and clinical studies, would be useful to understand more about the demineralization resistance and remineralization effects of these fluoride-releasing restorative materials.

## 5. Conclusions

Within the limitations of an in vitro study, our findings demonstrated that alkasite restorative material (CN) is a superior fluoride-releasing restorative material in stimulating demineralization and remineralization in an enamel model compared to the FI and RL groups. Hence, alkasite restorative material can be an effective material when used in cariogenic environments. However, to evaluate subsurface enamel lesions and to confirm these findings, additional methods such as animal or clinical studies may be useful.

## Figures and Tables

**Figure 1 materials-14-04554-f001:**
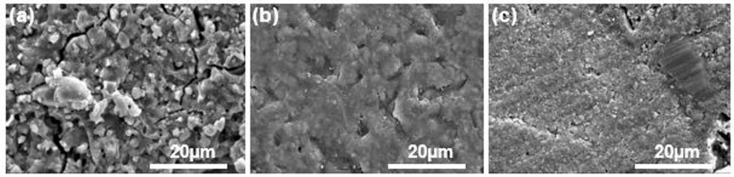
Scanning electron microscopy (SEM) images at 2000× of the surfaces of the materials used in this study: FI group (**a**), RL group (**b**), and CN group (**c**).

**Figure 2 materials-14-04554-f002:**
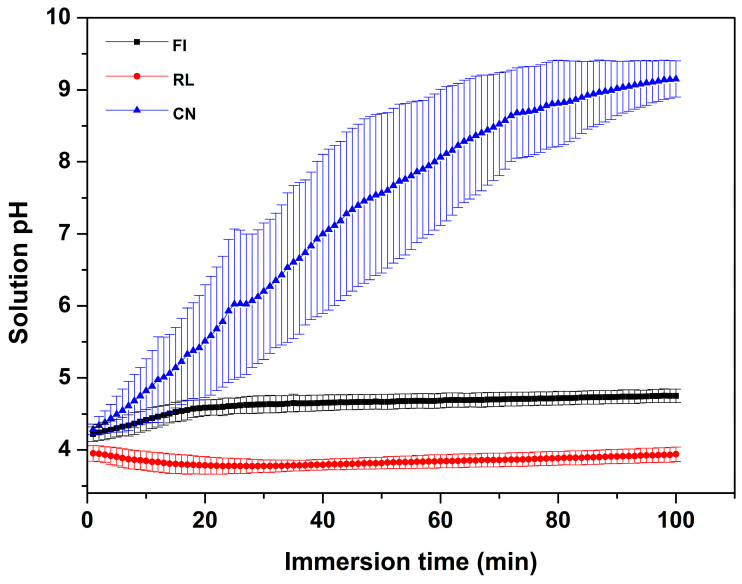
pH change in lactic acid solution with each experimental material. Each value indicates the mean of 5 repeated measurements, and the error bars represent the deviation of the mean values (mean ± standard deviation; n = 5).

**Figure 3 materials-14-04554-f003:**
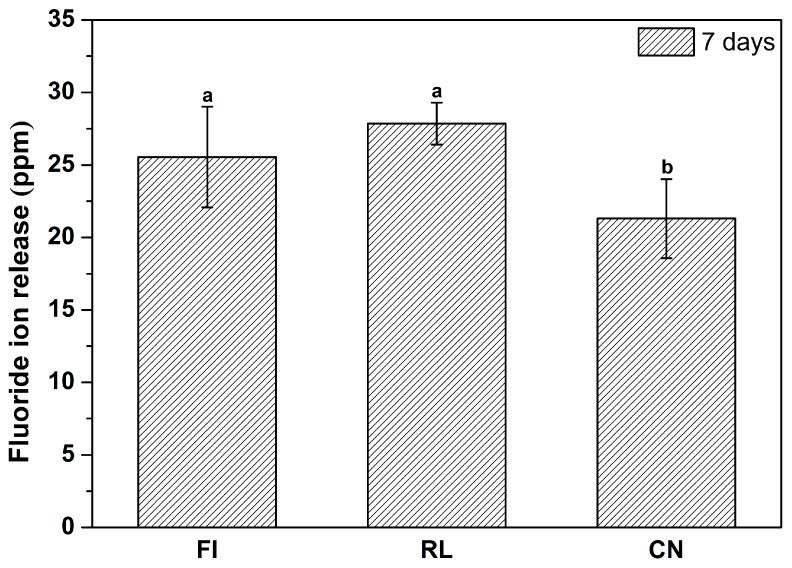
Fluoride ion release from the studied materials for 7 days. Each value indicates the mean of 6 measurements, and the error bars represent the standard deviation of the mean values (mean ± standard deviation; n = 6). Differences in lowercase letters above the bar graph indicate significant differences between each group (*p* < 0.05).

**Figure 4 materials-14-04554-f004:**
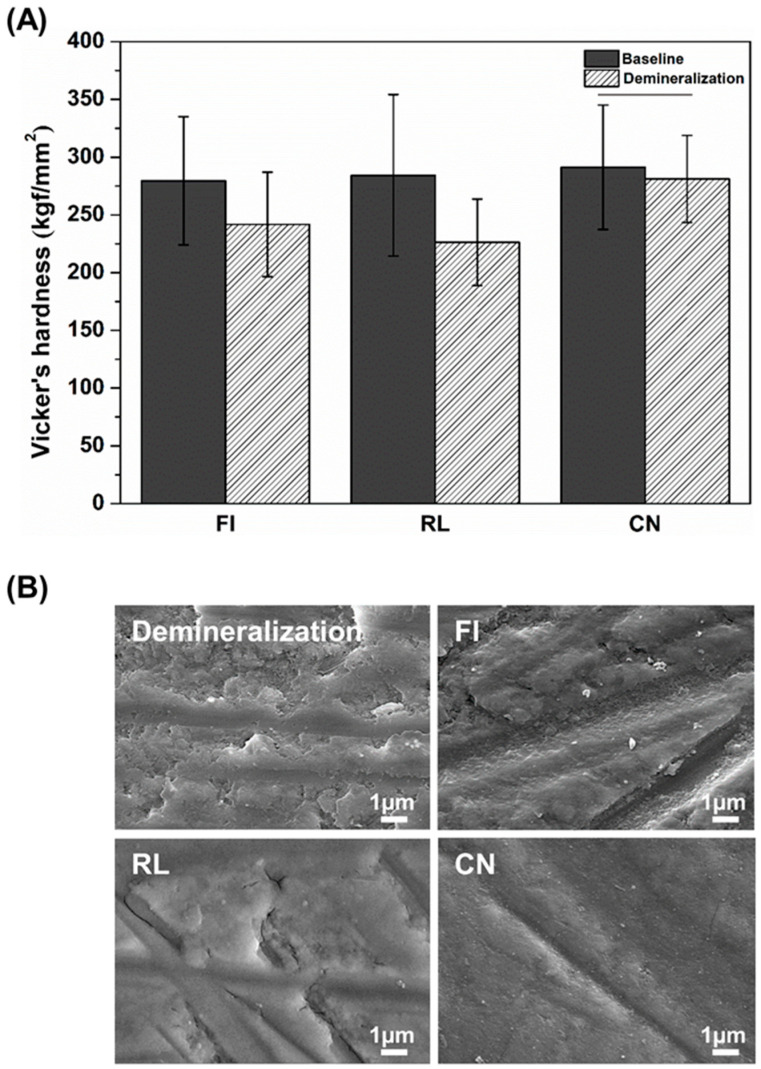
Microhardness of the enamel surface filled with each restorative material (**A**). Each value indicates the mean of 10 measurements, and the error bars represent the standard deviation of the (mean ± standard deviation; n = 10). Horizontal bar: no significant difference before and after immersion in demineralization (*p* > 0.05). Representative images of enamel surface after demineralization process (**B**).

**Figure 5 materials-14-04554-f005:**
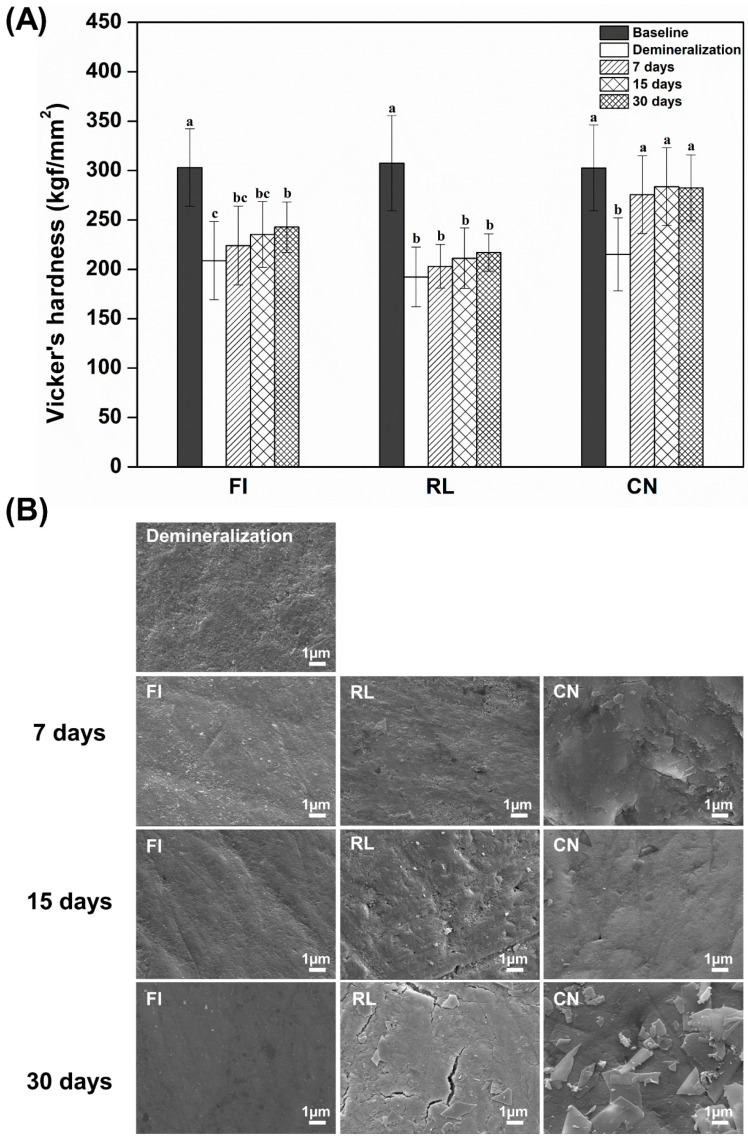
Microhardness of enamel surface filled with each restorative material (**A**). Each value indicates the mean of 10 measurements, and the error bars represent the standard deviation of the (mean ± standard deviation; n = 10). Differences in lowercase alphabetical letters above the bar graph indicate differences comparing periods for the same groups (*p* < 0.05). Representative images of enamel surface after the remineralization process (**B**).

**Table 1 materials-14-04554-t001:** Fluoride-releasing restorative materials investigated in this study.

Cord	Product	Manufacturer	Type	Composition
FI	Fuji IX	GC Co, Tokyo, Japan	Glass-ionomer cement	Alumino-fluoro-silicate glass polyacrylic acid, distilled water
RL	Riva Light Cure	SDI Limited, Bayswater, Victoria, Australia	Resin-modified glass-ionomer cement	Fluoro-alumino-silicate glass polyacrylic acid, tartaric acid, HEMA, camphorquinone
CN	Cention N	Ivoclar Vivadent, Schaan, Liechtenstein	Alkasite restorative material	Calcium fluorosilicate glass, Ba-Al silicate glass, Ca-Ba-Al fluorosilicate glass, ytterbium trifluoride, isofiller UDMA, DCP, aromatic aliphatic-UDMA

UDMA: urethane dimethacrylate, DCP: tricyclodecan-dimethanol dimethacrylate; Aromatic aliphatic-UDMA: tetramethyl-xylylen-diurethane dimethacrylate.

**Table 2 materials-14-04554-t002:** Ca ion concentration of each restorative material.

Material	Calcium Ion Concentration (ppm)		
	7 Days (Mean ± SD)	15 Days (Mean ± SD)	30 Days (Mean ± SD)
**FI**	0.20 ± 0.01 ^Aa^	0.18 ± 0.06 ^Aa^	0.19 ± 0.05 ^Aa^
**RL**	0.30 ± 0.12 ^Aa^	0.24 ± 0.07 ^Aa^	0.25 ± 0.05 ^Aa^
**CN**	32.51 ± 4.88 ^Ab^	32.32 ± 5.68 ^Ab^	34.06 ± 3.60 ^Ab^

The same uppercase letters indicate no statistically significant difference when comparing each period. The same lowercase letters indicate there were no statistically significant differences when comparing each period (*p* > 0.05).

## Data Availability

Not applicable.

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
