# Peer review of "Enamel Demineralization Resistance and Remineralization by Various Fluoride-Releasing Dental Restorative Materials"

_materials, 2021, doi:10.3390/ma14164554_

Round 1

Reviewer 1 Report

This is my personal review of the manuscript entitled "Enamel demineralization resistance and remineralization by various novel fluoride releasing dental restorative materials". The paper is proposed by Min-Ji Kim, Myung-Jin Lee, Kwang-Mahn Kim, Song-Yi Yang, Ji-Young Seo, Sung-Hwan Choi, and Jae-Sung Kwon for publication in the journal Materials (MDPI).

The research deals with the test of three commercial materials used for dentistry. Particularly, the work describes the way these materials may support the resistance against demineralization of tooth enamel and the remineralization of enamel by fluoride releasing. The surface of these materials is characterized by scanning electron microscopy. Their behavior in solution is assessed by pH measurements and by determining the fluoride and calcium ions concentrations after several days of immersion. At last, the microhardness of the three materials is studied before and after the immersion tests.

In my opinion the paper contains some interesting results but there is no real discussion of the results. The paper is mainly a technical report of several experiments. To make it publishable, the paper should answer to following questions:

  - What is the novelty of the study? What does it bring to the readers to motivate some new experiments?

  - What are the mechanisms involved in the observed results? The results should be deeply discussed and not just presented as “similar to other previous results”. What is the novelty of this work? Why?

  - If roughness is an important parameter, it should be measured.

  - The end of the paper indicates that the results have limitations in evaluating the enamel remineralization. So, I am not convinced by the originality of the study described in the last sentence of the discussion section.

  - There is no conclusion but just an indication that the results are insufficient and some more characterizations are necessary.

For these reasons, I consider that this article is not suitable for publication in the journal Materials (MDPI). I recommend rejection.

Author Response

Reviewer #1  

Q 1)

What is the novelty of the study? What does it bring to the readers to motivate some new experiments?

A 1)

Thank you for your thoughtful comments on the manuscript. Your valuable comments have assisted me to think about and add information on the originality in this research.

The newly introduced restorative material known as an “alkasite” are the dual-cured resin-based material containing alkaline glass where there has been lack of information on comparing properties such as resistance and remineralization by these materials with conventional products. Here, Cention N (CN) was considered as one of an alkasite material,  where study was conducted simulating the acid environment by applying directly the materials on the artificial cavity and demineralization and remineralization effects were observed simultaneously. Such information is now added in Introduction part.

Q 2)

What are the mechanisms involved in the observed results? The results should be deeply discussed and not just presented as “similar to other previous results”. What is the novelty of this work? Why?

A 2)

Sorry for the lack of explanation in Discussion. According to your suggestion, we have now added following sentence related to possible mechanisms as follows in Discussion;

4. Discussion

Na+ is one of the main ions enable to exchanging process with protons (H+) from the solution, the Na2O content determines the alkalizing ability of the filler. By containing such a alkaline glass, the CN group has a capacity for rapid acid neutralization, which confirms the effect of demineralization resistance when the restorative is applied to the enamel. In addition, fluoride ion releasing fluoroapatite, which has more effective demineralization resistance than hydroxyapatite. Consequently, CN is shown to have superior demineralization resistance than other two groups by simultaneously releasing various ions.

CN releases various ions such as Ca2+, Na+ and F- that it increases the pH and forms apatite when immersing the artificial saliva. These results are supposed to be affected by forming apatite through ion releasing.

Q 3)

If roughness is important parameter, it should be measured.

A 3)

We concur that roughness is a major factor for evaluating the surface of enamel. However, we did not carry out such analyses as the existing research can be interpreted that would provide enough information along with results of SEM images that are included in this study. We have now added more information with following references regard to evaluating surface with SEM and previous studies related to roughness.

Dong, Z.; Chang.; Zhou, Y.; Lin, K. In vitro remineralization of human dental enamel by bioactive glasses. Journal of materials science 2011, 461 1591-1596.

Q 4)

The end of the paper indicates the results have limitations in evaluating the enamel remineralization. So, I am not convinced by the originality of the study described in the last sentence of the discussion section.

A 4)

Sorry for the confusion. According to your comment, we now have added and modified sentences in end of Discussion as follow;

4. Discussion

To verify the caries progression exactly, SEM images and microhardness of enamel surface have limitation in evaluating enamel subsurface lesions enamel. However, this measurement is useful for evaluating initial structural change according to acid exposure and applying materials.

Q 5)

There is no conclusion but just an indication that the results are insufficient and some more characterizations are necessary.

A 5)

Again, sorry for the confusion and insufficient information in conclusion.

According to your suggestion, we have now modified the conclusion clearly as follow:

5. Conclusions

Within the limitations of in vitro study, this study demonstrated that alkasite restorative material (CN) is a superior fluoride releasing restorative material in simulating demineralization and remineralization on enamel model compared with FI and RL groups. However, to evaluate subsurface enamel lesions, additional methods such as confocal laser scanning electron is needed.

Reviewer 2 Report

The article is about the investigation of resistance against the demineralization and remineralization on enamel surface by various fluoride releasing restorative materials. However, some changes are needed:

  1. The introduction must to be rewritten with updated references from 2021.
  2. Line 160 ...: which are the importance of the comments and observations about figure 1? There are scientifically relevant?
  3. Line 181, line 192: since are no significant differences, I think also the producers knows and there are no explanations about the scientific point of view: useless!
  4. The differences between those three materials which are mentioned in Discussion section are irelevevant.
  5. The study seems to be original, but from scientific point of view I cannot see relevant conclusions for other researchers.
  6. Conclusions are too superficial and are not reflecting the results from the paper. It is very hard to understand the scientific soundess of the article, since the authors are comparing three different materials and there are no conclusions to show at the end of the paper. Moreover, the authors are not specifying WHY they compare those materials!?

If the authors rewrite entirely the article there can be some chances to be suitable for publication!

Author Response

Reviewer #2  

Q 1)

The introduction must to be rewritten with updated references from 2021.

A 1)

Thank you for your advice on the manuscript. According to your comment, we now have added references from 2021.

Q 2)

Line 160 …: which are the importance of the comments and observations about figure 1? There are scientifically relevant?

A 2)

The observation was conducted to reveal the thin and irregular pieces shown in SEM images (Figure 5) of CN group. We supposed to be influenced by the CN itself when observing the material surface and such information is now included.

Q 3)

Line 180, line 192: since are no significant differences, I think also the producers knows and there are no explanations about the scientific point of view: use less!

A 3)

Sorry for the lack of information related to scientific point of view. We now have modified Discussion substantially including information such as below;

4. Discussion

In addition, fluoride ion releasing forms fluoroapatite, which has more effective demineralization resistance than hydroxyapatite. CN releases various ions such as Ca2+, Na+ and F- that it increases pH and forms apatite when immersing the artificial saliva.

Q 4)

The differences between those three materials which are mentioned in Discussion section are irelevevant.

A 4)

Again, apologies for including some of information that may not be relevant to this study. Discussion is now substantially modified with more relevant information.

Q 5)

The study seems to be original, but from scientific point of view I cannot see relevant conclusions for other researchers.

A 5)

According to your comment, we have added to the scientific point of view in Introduction, Discussion and Conclusion. Also, possible mechanisms were considered with sentences in both Discussion and Conclusion as below:

4. Discussion

Also, Na+ is one of the main ions enable to exchanging process with protons (H+) from the solution, the Na2O content determines the alkalizing ability of the filler.  By containing such a alkaline glass, the CN group has a capacity for rapid acid neutralization, which confirms the effect of demineralization resistance when the restorative is applied to the enamel.

5. Conclusion

Within the limitations of in vitro study, this study demonstrated that alkasite restorative material (CN) is a superior fluoride releasing restorative material in simulating demineralization and remineralization on enamel model compared with FI and RL groups.  However, to evaluate subsurface enamel lesions, additional methods such as confocal laser scanning electron is needed.

Q 6)

Conclusions are too superficial and are not reflecting the results from the paper. It is very hard to understand the scientific soundess of the article, since the authors are comparing three different materials and there are no conclusions to show at the end of the paper. Moreover, the authors are not specifying WHY they compare those materials!?

A 6)

Thank you for your thoughtful comments on the manuscript. Your valuable comments have assisted me to think about the purpose of comparing those three materials in this research. Therefore, it is added to the introduction as below:

In conclusion, CN has alkaline glass filler embedded into resin matrix and release fluoride ion, this feature simultaneously contains that previously mentioned two materials. Therefore, we have compared those three fluoride releasing restorative mateirals.

Round 2

Reviewer 1 Report

In my opinion, the proposed modifications increase the quality of the paper.

The results are interesting and appropriately presented.

However, I don't find any good description of the novelty of the paper. The discussion should be deeply improved to provide ideas to develop some new experiments.

I consider that some corrections are still necessary but the paper can be published for these interesting new results.

Author Response

Comment:

In my opinion, the proposed modifications increase the quality of the paper.

The results are interesting and appropriately presented.

However, I don't find any good description of the novelty of the paper. The discussion should be deeply improved to provide ideas to develop some new experiments.

I consider that some corrections are still necessary but the paper can be published for these interesting new results.

Response:

Thank  you very much for your review of this paper. Now, we have added details in discussions to highlight novelty of this paper. Also entire manuscript is now proofread for English grammar for clarification of the paper.

Reviewer 2 Report

The introduction and conclusion sections still needs some improvement.

Author Response

Comment:

The introduction and conclusion sections still needs some improvement.

Response:

Thank  you very much for your review of this paper. Now, we have added details in both introduction and conclusion of this paper. Also entire manuscript is now proofread for English grammar for clarification of the paper.